# Information and Advance Care Directives for End-of-Life Residents with and without Dementia in Nursing Homes

**DOI:** 10.3390/healthcare11030353

**Published:** 2023-01-26

**Authors:** Emilio Mota-Romero, Olga Rodríguez-Landero, Rocío Moya-Dieguez, Glaucione Marisol Cano-Garzón, Rafael Montoya-Juárez, Daniel Puente-Fernández

**Affiliations:** 1Andalusian Health Service District Metropolitano Granada, Salvador Caballero Primary Care Centre, Institute for Biosanitary Research of Granada (IBS), University of Granada, 18014 Granada, Spain; 2Virgen de las Nieves University Hospital, 18014 Granada, Spain; 3Entreálamos Gerontological Centre, Atarfe, 18230 Granada, Spain; 4María Zayas Nursing Home, Belicena, 18101 Granada, Spain; 5Department of Nursing, Institute for Biosanitary Research of Granada (IBS), University of Granada, 18014 Granada, Spain

**Keywords:** nursing home, dementia, decision making, advance care planning

## Abstract

Background: Communication and advance care directives may be affected by the presence of dementia. We sought to describe the information and end-of-life preferences provided to nursing homes residents and their families. Methods: Trained nurses collected information from 124 residents randomly selected with palliative care needs from eight nursing homes. Results: A total of 54.4% of the residents with dementia had been provided with information about their state of health, compared to 92.5% of the residents without dementia (*p* < 0.01); family members exhibited no differences regarding information (*p* = 0.658), regardless of whether the resident was cognitively impaired. Most advance care interventions remained unexplored, except for cases where a transfer to hospital (81.5%) or serotherapy (69.4%) was desired. Decisions regarding palliative sedation (*p* = 0.017) and blood transfusion (*p* = 0.019) were lower among residents with dementia. Conclusions: Residents, especially residents with dementia, are provided with limited information and their preferences are inadequately explored.

## 1. Introduction

According to Eurostat, 3.8% of women and 1.9% of men over 65 years of age reside in nursing homes [1]. Nursing homes are likely to become increasingly important as a place of residence when the patient begins to require continuous and complex care [2]. Elderly people living in nursing homes tend to be highly frail due to chronic comorbidities, dementia, and dependency as well as high palliative care needs [3].

Palliative care has traditionally been aimed at cancer patients and health care professionals receive insufficient training in managing comorbidities in this age group [4]. Furthermore, another factor that may lead to reduced access to palliative care for the elderly is the lack of integration between nursing homes and health systems in different countries [5]. Institutions such as the World Health Organization recommend the use of strategies such as palliative care among people over 65 years of age, and particularly in nursing homes [4]. 

Palliative care is interdisciplinary care that focuses on improving the quality of life of individuals of any age living with a life-threatening illness, as well as that of their families [6]. This approach includes, prevention, early identification, comprehensive assessment, and management of physical issues, such as pain and other distressing symptoms, psychological distress, spiritual distress, and social needs. 

Palliative care provides support to help patients live as fully as possible until death by facilitating effective communication, helping them, and their families determine goals and preferences of care through advance care planning (ACP).

ACP is a care relationship process involving an individual and the health care professionals involved in their care, whereby they discuss the values, wishes, and preferences to be taken into consideration in order to make joint decisions about the care and interventions that the individual will receive as a resident, especially when they are at the end of their life and are no longer able to give their consent [7]. In order to successfully plan for this, it is essential to ensure continuous, reciprocal, unhindered communication between professionals, residents, and families.

Many individuals come to their end-of-life process without any planning, which increases the likelihood of them receiving more aggressive life-extending therapeutic interventions [8]. Potential factors hindering ACP in nursing homes include unwillingness to talk about death, work overload, and lack of coordination between these facilities and the emergency services [9,10].

In Spain, even though specific legislation on PCAs has been enacted over the last 20 years at the national level, the number of people expressing their wishes about their future care through PCAs is still very low [11]. According to the Spanish Government, the average percentage of completion for the Spanish total population was 0.82% in 2022 [12]. Although there is a lack of specific literature about ACP, some studies indicate that there is a low participation of residents of nursing homes in decision making about their health [13,14]. Blanca-Gutiérrez, Grande-Gascón, & Linares-Abad [13] highlighted in a previous study that residents in nursing homes give up participation in decision making in exchange for care, safety, and protection. Furthermore, professionals are reluctant to promote ACP among nursing homes residents, due to death taboo, lack of training and lack of institutional support among other factors [14].

ACP plays an important role for cognitively impaired individuals, as it reduces the family’s feelings of guilt and fear of failing to comply with the resident’s preferences [15]. Although strategies for ACP have to date focused primarily on families [16], individuals with mild dementia are increasingly becoming involved [17].

As has been highlighted in other studies [3], there is a high prevalence of cognitive impairment and dementia in Spanish nursing homes, so it is relevant to study how ACP is developed in this specific context. 

The objective of this study is to describe the information made available to residents and their families in nursing homes in Spain and their preferences regarding decision making and the information they wish to receive, and to ascertain the presence of any differences in information provision and ACP between residents with and without dementia living in nursing homes.

## 2. Materials and Methods

### 2.1. Design

Cross-sectional descriptive design.

### 2.2. Settings and Participants

Eight nursing homes in Andalusia (Spain) were selected using a convenience sampling method based on their institutional characteristics (presence of a multi-disciplinary team, the potential involvement of professionals, and the presence of both public and private beds). All the centers included in this study had more than 60 beds. In Andalusia, nursing homes with more than 60 beds are required to offer 24 h nursing services and their own medical care [18].

For a population of 552 residents, a minimum sample of 110 cases was estimated to detect a mean difference of 0.70 (SD 1), for a p value less than 0.05 and a power of 80%. 

A trained nursing home nurse randomly recruited 17 residents, according to a previous study that describe nursing homes disease distribution [19]; 40% with dementia, 40% with specific organ failure (chronic heart disease, COPD, chronic kidney disease and chronic liver disease) and 20% with cancer. Palliative care needs corroborated with the NECPAL CCOMS-ICO^©^ tool [20]. The NECPAL main objective is to identify persons with palliative care needs and life-limiting prognosis in health and social services, and it evaluates 13 multidimensional parameters. Residents in their final days, residents staying temporarily at the nursing home and residents without any family contacts were excluded. More details of this sample are shown in previous studies [21,22]. 

### 2.3. Data Collection Procedure

Data collection took place between June 2019 and January 2020. All residents (or their proxies in case of dementia) were duly informed about this study and signed an informed consent form.

A restricted-access online platform was used to administer all study questionnaires: sex, age, conditions, comorbidities, Barthel Index [23], and an ad hoc questionnaire to assess how well they were informed, their information preferences, the communication of diagnosis, treatment, symptoms, prognosis, the presence of conspiracy of silence, proxy decision makers, and advance care directives. Information was gathered directly from residents, family members and resident medical charts. Conspiracy of silence was understood as that situation in which the professional and family members knew the prognosis of the disease, but due to family pressures, agreement was reached not to alert the resident about the proximity of the end of his/her life [24]. They were also asked whether decisions had been made about specific interventions included in the Spanish Ministry of Health’s advance care directive document. 

### 2.4. Data Analysis

Chi-squared and Mann–Whitney’s U tests were used to explore the differences regarding information needs, preferences, and ACP between residents with and without dementia. 

Regarding information needs, the responses were grouped into two values based on whether or not the nurses had provided information to the resident. For information preferences, responses were grouped into two values according to whether or not residents had expressed any preferences. Concerning specific interventions, responses were grouped into two values depending on whether or not a decision had been made to carry out the intervention, including the cases in which it was not explored in the latter value. The statistical significance threshold for all tests was set at 0.05.

This study was approved by the Research Ethics Committee for [blinded for review] (0706-N-17).

## 3. Results

From an eligible population of 552 residents, 149 residents were selected, but only 124 were analyzed for this study. A total of 82 were women (66.1%) with a mean age of 84.7 years (SD = 8.12). Barthel index mean was 49.76 points (SD = 32.38), 46% had dementia, 40.3% had chronic heart disease, and 25.8% had COPD.

A total of 62 residents (50%) were fully informed about their health status. This percentage was lower in the case of residents with dementia (31.6%, n = 18). By contrast, 92.7% of all families reported being fully informed about the resident’s state of health.

A total of 44% of all residents (n = 55) and 21.1% (n = 12) of residents with dementia expressed a desire to be informed about their health status. Regarding families, 87.9% stated that they would like to be informed. Table 1 shows resident/family information level and preferences.

Residents with dementia received a lower proportion of information about their health status (full or partial), compared to residents without dementia (54.4% vs. 92.5%; *p* < 0.01). The percentage of families who received information about the resident’s state of health was similar between residents with and without dementia (98.2% vs. 97.0%; *p* = 0.657).

Residents with dementia expressed their preferences regarding their desire to be informed in a lower proportion than those without dementia (21.1% vs. 64.2%; *p* < 0.01). By contrast, the percentages of families expressing their preferences were similar in both groups (93% vs. 85.1%; *p* = 0.166).

The most frequently communicated aspects were diagnosis (99.2%) and treatment (99.2%), and the least frequently communicated was prognosis (84.7%). No differences were observed between residents with and without dementia. Conspiracy of silence remained unexplored in 74.2% of cases. In the cases where conspiracy of silence was explored (n = 32), its presence was confirmed in 11 (34.4%). Table 2 shows the specific information held by the resident or family.

Although nurses confirmed that treatment was agreed with the resident/family in most cases (95.9%), only a few cases had an appointed proxy decision maker (55.6%) and even fewer cases had advanced care directives in place (1.6%). The percentage of cases with an appointed proxy was similar in residents with and without dementia (54.4% vs. 56.7%; *p* = 0.795).

Regarding ACP, potential interventions were not explored with the resident or family or no decision was made in many cases. Figure 1 shows the decision making for specific interventions. The most explored interventions were use of serotherapy and transfer to hospital. In both cases, a high percentage of residents/families agreed to these interventions being carried out (69.4% and 81.5%, respectively).

The percentage of residents with dementia for whom a decision had been made regarding blood transfusion (*p* = 0.019) and palliative sedation (*p* = 0.017) when appropriate was lower than the percentage of residents without dementia.

## 4. Discussion

Our results suggest that elderly people living in nursing homes in Spain are poorly informed about their state of health. Results are even poorer for residents with dementia. Half of the residents of the sample had an identified family proxy who might participate in decision making. The percentage of residents with advance directives was very low. Most of the potential interventions were not discussed with families or residents. The most discussed interventions were serotherapy and transfer to hospital. 

Among the most striking results of this study is that only 50% of all residents had been fully informed about their health status. This percentage was higher in cognitive intact residents (65%) and lower in the case of residents with dementia (31.6%). By contrast, a high percentage of families of our sample had been being fully informed about the residents’ state of health. This is congruent with the residents’ desire to be informed about their health status. As has been highlighted in the background section, previous studies, that indicate that there is a low participation of residents of nursing homes in decision making about their health [13,14].

Prognosis is the aspect that residents and families were least informed about, despite this information being essential for decision making. A life prognosis of less than 6 months is the factor that most influences the type of care that family members felt that residents with advanced dementia should receive [25]. Furthermore, a 6-month life prognosis is linked to a more palliative approach to care [26]. Nevertheless, specific instruments may be required for residents with dementia, whose needs for palliative care may differ from those of residents without this condition. 

As observed in the relevant literature in Spain [27], the percentages of cases where a proxy or advance care directives had been appointed were similarly low among residents with and without dementia. It is important to highlight that, in relation to advance directives, the average percentage of completion for the Spanish total population was 0.82% in 2022 [12]. Andreasen et al. [28] in a retrospective study as part of the European PACE Project, reported results similar to those observed in our study and in the Spanish population for the Italian population, where only 0.1% had advance directives compared to 76.9% of long-term care residents in the United Kingdom. This may be due to cultural aspects as pointed out by Meñaca et al. [29] in a comparison between Spain, Italy and Portugal, where they show similar results regarding advance directives.

A recent review [11] has studied the reasons why ACDs are not widely used in Spain by the general population, pointing out four aspects that are failing: (1) ACDs fail due to the lack of adequate training of health care professionals on end of life and ACDs, both conceptually and in the application of Spanish legislation; (2) lack of public awareness campaigns to raise knowledge about ACD among both patients and professionals; (3) the existence of very complex bureaucratic documentation and application procedures that are a barrier for both patients to register them and professionals to consult them when the time comes; (4) the persistence of a paternalistic medical culture, both among patients and health professionals, which complicates shared decision making with patients and their families. The appointment of proxies and the implementation of advance care directives should be promoted in nursing homes.

According to our results, most interventions, except for transfer to hospital and serotherapy, had remained unexplored. This is consistent with the low degree of implementation of ACP in these facilities [30]. Ottoboni et al. [31], in their study in 12 Italian nursing homes evaluating the knowledge and attitudes of professionals towards ACP pointed out that nursing home staff might bother residents and relatives when they introduce ACP, although they perceived this as a potential benefit for end-of-life coping. 

Reinhardt et al. [32] in their study of family members of residents with dementia in the USA suggested that proper communication about end-of-life processes positively affects relatives’ confidence in making palliative care-related decisions. Therefore, it is necessary to discuss specific interventions at end of life in nursing homes. 

Our results suggest that there is a notable difference in the level of information between residents with and without dementia. However, they do not reveal any major differences in the exploration of ACP between cognitively impaired and non-cognitively impaired individuals. It is important not to exclude residents from decision making despite the difficulty that dementia poses when communicating with them [16]. 

According to a recent review [33], the degree to which persons with dementia were involved in planning their future care varies across the different settings, being more likely to participate in these decisions in the community than in acute settings or nursing homes. Dementia staging is essential in nursing homes to identify mild cognitive impaired residents and introduce ACP as soon as possible and always before residents lose their capacity to make decisions by themselves. It would be also inappropriate to try to force moderate and severe dementia residents to make medical decisions. In those cases, the designated surrogate would have to make all decisions for the resident, including health care issues.

The limitations of this study include the fact that no distinction was made between the decision making of residents with moderate or severe dementia, because cognitive impairment staging was not provided by nursing homes professionals for this study. Although the sample was randomized, it is unclear if the sample is representative of total palliative residents in nursing homes, as there is no previous bibliography in this sense in our context.

## 5. Conclusions

In summary, institutionalized elderly people are often excluded from information and decision making in their own life processes. Residents are significantly less well informed than their families, particularly in the case of residents with dementia. Half of the residents of the sample had an identified family proxy who might participate in decision making. The percentage of residents with advance directives was very low. Most of the potential interventions were not discussed with families or residents. The most discussed interventions were serotherapy and transfer to hospital. Few significant differences were found in ACP between residents with and without dementia. 

Protocols, interventions, and action plans need to be developed for residents with and without dementia to improve the provision of information and the recording of decision making in nursing homes.

## Figures and Tables

**Figure 1 healthcare-11-00353-f001:**
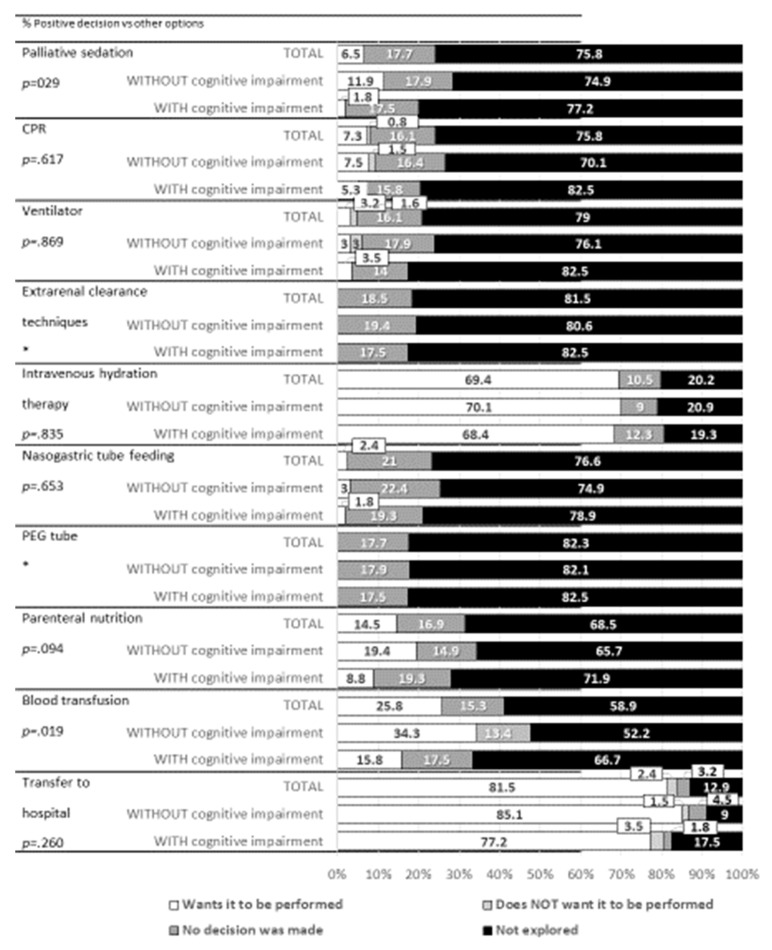
Decision making on specific interventions. * No analysis was possible due to a lack of cases.

**Table 1 healthcare-11-00353-t001:** Resident/family information levels and preferences.

		Total n = 124	With Cognitive Impairment n = 57		Without Cognitive Impairment n = 67		*p*
			n (%)				
Resident	Fully informed	62 (50.0%)	18 (31.6%)	31 (54.4%)	44 (65.7%)	62 (92.5%)	0.000 *
	Partially informed	31 (25.0%)	13 (22.8%)		18 (26.9%)		
	No information	16 (12.9%)	16 (28.1%)	26 (45.6%)	0 (0.0%)	5 (7.5%)	
	Not explored	15 (12.1%)	10 (17.5%)		5 (7.5%)		
Family	Fully informed	115 (92.7%)	54 (94.7%)	56 (98.3%)	61 (91.0%)	65 (97%)	0.658 *
	Partially informed	6 (4.8%)	2 (3.5%)		4 (6.0%)		
	No information	1 (0.8%)	0 (0.0%)	1 (1.8%)	1 (1.5%)	2 (3%)	
	Not explored	2 (1.6%)	1 (1.8%)		1 (1.5%)		
Resident preferences	Wants to be informed	55 (44.4%)	12 (21.1%)	12 (21.1%)	43 (64.2%)	43 (64.2%)	0.000 **
	Does not want to be informed	0 (0.0%)	0 (0.0%)		0 (0.0%)		
	No opinion	49 (39.5%)	33 (57.9%)	45 (78.9%)	16 (23.9%)	24 (35.8%)	
	Not explored	20 (16.1%)	12 (21.1%)		8 (11.9%)		
Family preferences	Wants to be informed	109 (87.9%)	53 (93.0%)	53 (93.0%)	56 (83.6%)	57 (85.1%)	0.167 **
	Does not want to be informed	1 (0.8%)	0 (0.0%)		1 (1.5%)		
	No opinion	2 (1.6%)	1 (1.8%)	4 (7.1%)	1 (1.5%)	10 (14.9%)	
	Not explored	12 (9.7%)	3 (5.3%)		9 (13.4%)		

* Fully/Partially informed vs. No information/Not explored; ** Wants to be informed/Does NOT want to be informed vs. No opinion/Not explored.

**Table 2 healthcare-11-00353-t002:** Specific information held by the resident or family.

Information	Total n = 124		With Cognitive Impairment n = 57		Without Cognitive Impairment n = 67		*p*
	Yes n (%)	No n (%)	Yes n (%)	No n (%)	Yes n (%)	No n (%)	
Diagnosis	123 (99.2%)	1 (0.8%)	56 (98.2%)	1 (1.8%)	67 (100.0%)	0 (0.0%)	0.276
Treatment	123 (99.2%)	1 (0.8%)	56 (98.2%)	1 (1.8%)	67 (100.0%)	0 (0.0%)	0.276
Symptoms	119 (96.0%)	5 (4.0%)	56 (98.2%)	1 (1.8%)	63 (94.0%)	4 (6.0%)	0.234
Prognosis	105 (84.7%)	19 (15.3%)	51 (89.5%)	6 (10.5%)	54 (80.6%)	13 (19.4%)	0.171
Collusion of silence is explored	32 (25.8%)	92 (74.2%)	14 (24.6%)	43 (75.4%)	18 (26.9%)	49 (73.1%)	0.770
Understands the clinical situation	120 (97.6%)	3 (2.4%)	54 (96.4%)	2 (3.6%)	66 (98.5%)	1 (1.5%)	0.457
Accepts the clinical situation	115 (93.5%)	8 (6.5%)	53 (94.6%)	3 (5.4%)	62 (92.5%)	5 (7.5%)	0.637
Treatment has been agreed	118 (95.9%)	5 (4.1%)	54 (96.4%)	2 (3.6%)	64 (95.5%)	3 (4.5%)	0.800
Has an appointed representative	69 (55.6%)	55 (36.9%)	31 (54.4%)	26 (45.6%)	38 (56.7%)	29 (43.3%)	0.795
Has advance care directives on record	2 (1.6%)	122 (98.4%)	1 (1.8%)	56 (98.2%)	1 (1.5%)	66 (98.5%)	*

* The statistic cannot be calculated due to the low number of cases.

## Data Availability

The data presented in this study are available upon request from the corresponding author. The data are not publicly available due to privacy restrictions.

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
