# Peer review of "Information and Advance Care Directives for End-of-Life Residents with and without Dementia in Nursing Homes"

_healthcare, 2023, doi:10.3390/healthcare11030353_

Round 1

Reviewer 1 Report

An interesting study with practical relevance.

In lines 63-64, authors state that "two nurses(...) randomly selected residents with palliative care needs based on the NECPAL CCOMS-ICO" - How was this randomization process carried out? Concerning the NECPAL instrument, it makes sense to present a brief explanation for a better understanding (even having placed, and well, a reference for better exploration).

Table 1 - In one place, they say residents, and in the other patients - change to residents, as shown in the rest of the document.

In line 147, the authors refer "Elderly people living in nursing homes in Spain are poorly informed about their state of health, in contrast to their families." - This makes sense to me to be mentioned in those with dementia. In those who don't have it, if I understood correctly, those who have information (total or partial are 92.5%), are not much different from the 97% relative to the family. The interpretation may be different if you talk about being fully informed, but this must be reflected in their statement.

Another detail, the study compared users with and without dementia and between family members of users with and without dementia. The study would benefit from the further analysis between groups. In other words, is this difference between residents with dementia and their relatives statistically significant?

The discussion, although brief, is focused on the findings. You could explore more relevant results.

The authors refer to a limitation, and it is well-identified. However, it seems to me that it would be necessary to mention whether the sample used is representative of the population and, if not, what sample would be required to generalize the results.

Author Response

Reviewer (R): An interesting study with practical relevance.

Authors (A): Thank you very much for your feedback. We hope that our responses and changes will improve the quality of the article.

(R): In lines 63-64, authors state that "two nurses(...) randomly selected residents with palliative care needs based on the NECPAL CCOMS-ICO" - How was this randomization process carried out? Concerning the NECPAL instrument, it makes sense to present a brief explanation for a better understanding (even having placed, and well, a reference for better exploration).

(A): The sample selection process has been clarified (Line 62):

Eight nursing homes in Andalusia (Spain) were selected using a convenience sampling method based on their institutional characteristics (presence of a multidisciplinary team, the potential involvement of professionals, and the presence of both public and private beds). All the centers included in the study had more than 60 beds. In Andalusia, nursing homes with more than 60 beds are required to offer 24-h nursing services and their own medical care [10]

A trained nursing home nurse randomly recruited 17 residents, according to a previous study that describe nursing homes disease distribution [11]; 40% with dementia, 40% with specific organ failure (chronic heart disease, COPD, chronic kidney dis-ease and chronic liver disease) and 20% with cancer. Palliative care needs were corroborated with NECPAL CCOMS-ICO© tool [12].

A brief explanation of the NECPAL tool has been included (Line 74)

The NECPAL main objective is to identify persons with palliative care needs and life-limiting prognosis in health and social services, and it evaluate 13 multidimensional parameters.

References have been added for further details (Line 77)

Further details of this sample are shown in previous studies [13, 14].

(R): Table 1 - In one place, they say residents, and in the other patients - change to residents, as shown in the rest of the document.

(A): According to your suggestion patient has been replaced by resident.

(R): In line 147, the authors refer "Elderly people living in nursing homes in Spain are poorly informed about their state of health, in contrast to their families." - This makes sense to me to be mentioned in those with dementia. In those who don't have it, if I understood correctly, those who have information (total or partial are 92.5%), are not much different from the 97% relative to the family. The interpretation may be different if you talk about being fully informed, but this must be reflected in their statement.

(A): Thanks for your suggestions. The above-mentioned phrase has been modified to avoid misunderstanding (Line 157).

Elderly people living in nursing homes in Spain are poorly informed about their state of health.

(R): Another detail, the study compared users with and without dementia and between family members of users with and without dementia. The study would benefit from the further analysis between groups. In other words, is this difference between residents with dementia and their relatives statistically significant?

(A): Figure 1 indicates which decisions had been made and which had not been explored, regardless of whether it was the patient or the relative who had made the decision. Unfortunately, we have no information about who made these decisions so we cannot offer further explanations in that sense.

(R): The discussion, although brief, is focused on the findings. You could explore more relevant results.

(A): Many thanks. New lines have been added to discussion section (Lines 167, 173 and 184)

Line 167: In relation to advance directives, the average percentage of completion for Spanish total population was 0.82% in 2022 [18].

Line 172: Ottoboni et al. [22] pointed out that nursing home staff might bother residents and relatives when they introduce ACP, although they perceived this as a potential benefit for end-of-life coping. 

Line 184: According to a recent review [24], the degree to which persons with dementia were involved in planning their future care varied across the different settings. Being more likely to participate in these decisions in the community than in acute settings or nursing homes. Dementia staging is essential in nursing homes to identify mild cognitive impaired residents and get ACP as soon as possible and always before residents lose their capacity to make decisions by themselves. It would be also inappropriate to try to force moderate and severe dementia residents to make medical decisions. In those cases, the designated surrogate would have to make all decisions for the resident, including health care issues.   

(R): The authors refer to a limitation, and it is well-identified. However, it seems to me that it would be necessary to mention whether the sample used is representative of the population and, if not, what sample would be required to generalize the results.

(A): A sample size estimation has been added to methods section (Line 68)

For a population of 552 residents a minimum sample of 110 cases was estimated to detect a mean difference of 0.70 (SD 1), for a p value less than 0.05 and a power of 80%.

And a new limitation has been added (Line 195)

Although sample was randomized, it is unclear if the sample is representative of total palliative patients in nursing homes, due there no previous bibliography in these sense in our context.

Reviewer 2 Report

Dementia STAGING is essential in this context.  If a patient is a FAST Stage 7c, for example, giving them any information about end of life planning would be inappropriate.  In those cases, the designated surrogate would have to make all decisions for the patient, including health care issues.  What is important in these situations is getting advance care planning BEFORE the patient loses capacity.

Perhaps I missed it, but I do not see any discussion of capacity evaluation on these patients with dementia.  Again, if a patient does not have capacity, it would be inappropriate to try to force them to make medical decisions and the results you found would actually demonstrate appropriate health care standards.

This article really cannot be published until there is at least a discussion of the stage of dementia the cohort was in (if it was just one;  if it's a spectrum of dementia you have a lot of work to do) and how this affected the numbers of patients with advance care planning in place.

As always, these issues stress the need for everyone to have an advance care plan in place before they need it and before they lose capacity to make medical decisions.

Author Response

Reviewer (R): Dementia STAGING is essential in this context.  If a patient is a FAST Stage 7c, for example, giving them any information about end-of-life planning would be inappropriate.  In those cases, the designated surrogate would have to make all decisions for the patient, including health care issues.  What is important in these situations is getting advance care planning BEFORE the patient loses capacity.

Authors (A): As it has been pointed out in the limitations, no distinction was made between the decision-making of residents with moderate or severe dementia, because cognitive impairment staging was not provided by nursing homes professionals for this study. This has been clarified in limitations (Line 195):

The limitations of this study include the fact that no distinction was made between the decision-making of residents with moderate or severe dementia, because cognitive impairment staging was not provided by nursing homes professionals for this study.

(R): Perhaps I missed it, but I do not see any discussion of capacity evaluation on these patients with dementia.  Again, if a patient does not have capacity, it would be inappropriate to try to force them to make medical decisions and the results you found would demonstrate appropriate health care standards.

(A): We agree that residents with moderate or severe dementia are not capable to participate in end-of-life decisions. Results on Table 1 and 2 are congruent with this idea. Nevertheless, relatives of dementia patients must participate in end-of-life planning and our results highlight they don´t do it.

Furthermore, Figure 1 indicates which decisions had been made and which had not been explored, regardless of whether it was the patient or the relative who had made the decision. Unfortunately, we have no information about who made these decisions and when these decisions were made, so we cannot offer further explanations in that sense.

Nevertheless, a paragraph has been added to discuss this important question.

According to a recent review [24], the degree to which persons with dementia were involved in planning their future care varies across the different settings, being more likely to participate in these decisions in the community than in acute settings or nursing homes. Dementia staging is essential in nursing homes to identify mild cognitive impaired residents and introduce ACP as soon as possible and always before residents lose their capacity to make decisions by themselves. It would be also inappropriate to try to force moderate and severe dementia residents to make medical decisions. In those cases, the designated surrogate would have to make all decisions for the resident, including health care issues.   

(R): This article really cannot be published until there is at least a discussion of the stage of dementia the cohort was in (if it was just one; if it's a spectrum of dementia you have a lot of work to do) and how this affected the numbers of patients with advance care planning in place.

(A): Although this important limitation we consider that the results of this study describe that there is no culture of advanced care planning in nursing homes in Spain. This manuscript (if it is published) could encourage policy makers and managers to promote advanced care planning in these centers.

(R): As always, these issues stress the need for everyone to have an advance care plan in place before they need it and before they lose capacity to make medical decisions.

(A): Additional lines have been added to discussion section to reinforce the importance of ACP in nursing homes (Line 184).

Round 2

Reviewer 2 Report

Thank you for addressing our concerns

Author Response

Editor: Dear Authors, your manuscript has been well modified based on the reviewers' comments. I did notice a discrepancy between the purpose of the study as described in the main text and the abstract. Please update the abstract. 

Author: As recommended by the editor, we have updated the abstract based on the response to the reviewers' comments. 

The abstract reads as follows:

..."Methods: Trained nurses collected information from 124 residents randomly selected with palliative care needs from eight nursing homes, according to a previous study that describe diseases distribution. Results:..."